# Solving inverse problem of Markov chain with partial observations

**Tetsuro Morimura**
IBM Research - Tokyo
tetsuro@jp.ibm.com

**Takayuki Osogami**
IBM Research - Tokyo
osogami@jp.ibm.com

**Tsuyoshi Idé**
IBM T.J. Watson Research Center
tide@us.ibm.com

## Abstract

The Markov chain is a convenient tool to represent the dynamics of complex systems such as traffic and social systems, where probabilistic transition takes place between internal states. A Markov chain is characterized by initial-state probabilities and a state-transition probability matrix. In the traditional setting, a major goal is to study properties of a Markov chain when those probabilities are known. This paper tackles an inverse version of the problem: we find those probabilities from partial observations at a limited number of states. The observations include the frequency of visiting a state and the rate of reaching a state from another. Practical examples of this task include traffic monitoring systems in cities, where we need to infer the traffic volume on single link on a road network from a limited number of observation points. We formulate this task as a regularized optimization problem, which is efficiently solved using the notion of natural gradient. Using synthetic and real-world data sets including city traffic monitoring data, we demonstrate the effectiveness of our method.

## 1 Introduction

The Markov chain is a standard model for analyzing the dynamics of stochastic systems, including economic systems [29], traffic systems [11], social systems [12], and ecosystems [6]. There is a large body of the literature on the problem of analyzing the properties a Markov chain given its initial distribution and a matrix of transition probabilities [21, 26]. For example, there exist established methods for analyzing the stationary distribution and the mixing time of a Markov chain [23, 16]. In these traditional settings, the initial distribution and the transition-probability matrix are given a priori or directly estimated.

Unfortunately, it is often impractical to directly measure or estimate the parameters (i.e., the initial distribution and the transition-probability matrix) of the Markov chain that models a particular system under consideration. For example, one can analyze a traffic system [27, 24], including how the vehicles are distributed across a city, by modeling the dynamics of vehicles as a Markov chain [11]. It is, however, difficult to directly measure the fraction of the vehicles that turns right or left at every intersection.

The inverse problem of a Markov chain that we address in this paper is an inverse version of the traditional problem of analyzing a Markov chain with given input parameters. Namely, our goal is to estimate the parameters of a Markov chain from partial observations of the corresponding system. In the context of the traffic system, for example, we seek to find the parameters of a Markov chain, given the traffic volumes at stationary observation points and/or the rate of vehicles moving between

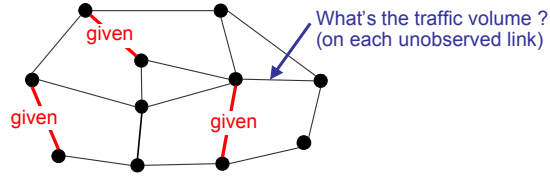

Figure 1: An inverse Markov chain problem. The traffic volume on every road is inferred from traffic volumes at limited observation points and/or the rates of vehicles transitioning between these points.

these points. Such statistics can be reliably estimated from observations with web-cameras [27], automatic number plate recognition devices [10], or radio-frequency identification (RFID) [25], whose availability is however limited to a small number of observation points in general (see Figure 1). By estimating the parameters of a Markov chain and analyzing its stationary probability, one can infer the traffic volumes at unobserved points.

The primary contribution of this paper is the first methodology for solving the inverse problem of a Markov chain when only the observation at a limited number of stationary observation points are given. Specifically, we assume that the frequency of visiting a state and/or the rate of reaching a state from another are given for a small number of states. We formulate the inverse problem of a Markov chain as a regularized optimization problem. Then we can efficiently find a solution to the inverse problem of a Markov chain based on the notion of natural gradient [3].

The inverse problem of a Markov chain has been addressed in the literature [9, 28, 31], but the existing methods assume that sample paths of the Markov chain are available. Related work of inverse reinforcement learning [20, 1, 32] also assumes that sample paths are available. In the context of the traffic system, the sample paths corresponds to probe-car data (i.e., sequence of GPS points). However, the probe-car data is expensive and rarely available in public. Even when it is available, it is often limited to vehicles of a particular type such as taxis or in a particular region. On the other hand, stationary observation data is often less expensive and more obtainable. For instance, web-camera images are available even in developing countries such as Kenya [2].

The rest of this paper is organized as follows. In Section 2, preliminaries are introduced. In Section 3, we formulate an inverse problem of a Markov chain as a regularized optimization problem. A method for efficiently solving the inverse problem of a Markov chain is proposed in Section 4. An example of implementation is provided in Section 5. Section 6 evaluates the proposed method with both artificial and real-world data sets including the one from traffic monitoring in a city.

## 2   Preliminaries

A discrete-time Markov chain [26, 21] is a stochastic process, $\boldsymbol{X} = (X_0, X_1, \dots)$, where $X_t$ is a random variable representing the state at time $t \in \mathbb{Z}_{\geq 0}$. A Markov chain is defined by the triplet $\{\mathcal{X}, p_\mathrm{I}, p_\mathrm{T}\}$, where $\mathcal{X} = \{1, \dots, |\mathcal{X}|\}$ is a finite set of states, where $|\mathcal{X}| \geq 2$ is the number of states. The function, $p_\mathrm{I} : \mathcal{X} \to [0, 1]$, specifies the initial-state probability, i.e., $p_\mathrm{I}(x) \triangleq \Pr(X_0 = x)$, and $p_\mathrm{T} : \mathcal{X} \times \mathcal{X} \to [0, 1]$ specifies the state transition probability from $x$ to $x'$, i.e., $p_\mathrm{T}(x' \mid x) \triangleq \Pr(X_{t+1} = x' \mid X_t = x)$, $\forall t \in \mathbb{Z}_{\geq 0}$. Note the state transition is conditionally independent of the past states given the current state, which is called the Markov property.

Any Markov chain can be converted into another Markov chain, called a Markov chain with restart, by modifying the transition probability. There, the initial-state probability stays unchanged, but the state transition probability is modified into $p$ such that

$$p(x' \mid x) \triangleq \beta p_\mathrm{T}(x' \mid x) + (1 - \beta) p_\mathrm{I}(x'), \tag{1}$$

where $\beta \in [0, 1)$ is a continuation rate of the Markov chain[1]. In the limit of $\beta \to 1$, this Markov chain with restart is equivalent to the original Markov chain. In the following, we refer to $p$ as the (total) transition probability, while $p_\mathrm{T}$ as a partial transition (or *p-transition*) probability.

Our main targeted applications are (massive) multi-agent systems such as traffic systems. So, restarting a chain means that an agent's origin of a trip is decided by the initial distribution, and the trip ends at each time-step with probability $1 - \beta$.

We model the initial probability and *p-transition* probability with parameters $\boldsymbol{\nu} \in \mathbb{R}^{d_1}$ and $\boldsymbol{\omega} \in \mathbb{R}^{d_2}$, respectively, where $d_1$ and $d_2$ are the numbers of those parameters. So we will denote those as $p_{I\nu}$ and $p_{T\omega}$, respectively, and the total transition probability as $p_\theta$, where $\boldsymbol{\theta}$ is the total model parameter, $\boldsymbol{\theta} \triangleq [\boldsymbol{\nu}^\top, \boldsymbol{\omega}^\top, \tilde{\beta}]^\top \in \mathbb{R}^d$ where $d = d_1 + d_2 + 1$ and $\tilde{\beta} \triangleq \varsigma^{-1}(\beta)$ with the inverse of sigmoid function $\varsigma^{-1}$. That is, Eq. (1) is rewritten as

$$p_\theta(x' \mid x) \triangleq \beta p_{T\omega}(x' \mid x) + (1 - \beta)p_{I\nu}(x'). \tag{2}$$

The Markov chain with restart can be represented as $M(\boldsymbol{\theta}) \triangleq \{\mathcal{X}, p_{I\nu}, p_{T\omega}, \beta\}$.

Also we make the following assumptions that are standard for the study of Markov chains and their variants [26, 7].

**Assumption 1** *The Markov chain $M(\boldsymbol{\theta})$ for any $\boldsymbol{\theta} \in \mathbb{R}^d$ is ergodic (irreducible and aperiodic).*

**Assumption 2** *The initial probability $p_{I\nu}$ and p-transition probability $p_{T\omega}$ are differentiable everywhere with respect to $\boldsymbol{\theta} \in \mathbb{R}^d$.*[2]

Under Assumption 1, there exists a unique stationary probability, $\pi_\theta(\cdot)$, which satisfies the balance equation:

$$\pi_\theta(x') = \textstyle\sum_{x \in \mathcal{X}} p(x' \mid x)\pi_\theta(x), \ \forall x' \in \mathcal{X}, \tag{3}$$

This stationary probability is equal to the limiting distribution and independent of the initial state: $\pi_\theta(x') = \lim_{t \to \infty} \Pr(X_t = x' \mid X_0 = x, M(\boldsymbol{\theta})), \ \forall x \in \mathcal{X}$. Assumption 2 indicates that the transition probability $p_\theta$ is also differentiable for any state pair $(x, x') \in \mathcal{X} \times \mathcal{X}$ with respect to any $\boldsymbol{\theta} \in \mathbb{R}^d$.

Finally we define hitting probabilities for a Markov chain of indefinite-horizon. The Markov chain is represented as $\tilde{M}(\boldsymbol{\theta}) = \{\mathcal{X}, p_{T\omega}, \beta\}$, which evolves according to the *p-transition* probability $p_{T\omega}$, not to $p_\theta$, and terminates with a probability $1 - \beta$ at every step. The hitting probability of a state $x'$ given $x$ is defined as

$$h_\theta(x' \mid x) \triangleq \Pr(x' \in \tilde{\boldsymbol{X}} \mid X_0 = x, \tilde{M}(\boldsymbol{\theta})), \tag{4}$$

where $\tilde{\boldsymbol{X}} = (\tilde{X}_0, \dots, \tilde{X}_T)$ is a sample path of $\tilde{M}(\boldsymbol{\theta})$ until the stopping time, $T$.

# 3 Inverse Markov Chain Problem

Here we formulate an inverse problem of the Markov chain $M(\boldsymbol{\theta})$. In the inverse problem, the model family $\mathcal{M} \in \{M(\boldsymbol{\theta}) \mid \boldsymbol{\theta} \in \mathbb{R}^d\}$, which may be subject to a transition structure as in the road network, is known or given a priori, but the model parameter $\boldsymbol{\theta}$ is unknown. In Section 3.1, we define inputs of the problem, which are associated with functions of the Markov chain. Objective functions for the inverse problem are discussed in Section 3.2.

## 3.1 Problem setting

The input and output of our inverse problem of the Markov chain is as follows.

- **Inputs** are the values measured at a portion of states $x \in \mathcal{X}_o$, where $\mathcal{X}_o \subset \mathcal{X}$ and usually $|\mathcal{X}_o| \ll |\mathcal{X}|$. The measured values include the frequency of visiting a state, $f(x)$, $x \in \mathcal{X}_o$. In addition, the rate of reaching a state from another, $g(x, x')$, might also be given for $(x, x') \in \mathcal{X}_o \times \mathcal{X}_o$, where $g(x, x)$ is equal to 1.

  In the context of traffic monitoring, $f(x)$ denotes the number of vehicles that went through an observation point, $x$; $g(x, x')$ denotes the number of vehicles that went through $x$ and $x'$ in this order divided by $f(x)$.

- **Output** is the estimated parameter $\boldsymbol{\theta}$ of the Markov chain $M(\boldsymbol{\theta})$, which specifies the total-transition probability function $p_\theta$ in Eq. (2).

The first step of our formulation is to relate $f$ and $g$ to the Markov chain. Specifically, we assume that the observed $f$ is proportional to the true stationary probability of the Markov chain:

$$\pi^*(x) = cf(x), \quad x \in \mathcal{X}_o, \tag{5}$$

where $c$ is an unknown constant to satisfy the normalization condition. We further assume that the observed reaching rate is equal to the true hitting probability of the Markov chain:

$$h^*(x' \mid x) = g(x, x'), \quad (x, x') \in \mathcal{X}_o \times \mathcal{X}_o. \tag{6}$$

## 3.2 Objective function

Our objective is to find the parameter $\boldsymbol{\theta}^*$ such that $\pi_{\theta*}$ and $h_{\theta*}$ well approximate $\pi^*$ and $h^*$ in Eqs. (5) and (6). We use the following objective function to be minimized,

$$L(\boldsymbol{\theta}) \triangleq \gamma L_d(\boldsymbol{\theta}) + (1 - \gamma)L_h(\boldsymbol{\theta}) + \lambda R(\boldsymbol{\theta}), \tag{7}$$

where $L_d$ and $L_h$ are cost functions with respect to the quality of the approximation of $\pi^*$ and $h^*$, respectively. These are specified in the following subsections. The function $R(\boldsymbol{\theta})$ is the regularization term of $\boldsymbol{\theta}$, such as $||\boldsymbol{\theta}||_2^2$ or $||\boldsymbol{\theta}||_1$. The parameters $\gamma \in [0, 1]$ and $\lambda \geq 0$ balance these cost functions and the regularization term, which will be optimized by cross-validation. Altogether, our problem is to find the parameter, $\boldsymbol{\theta}^* = \arg\min_{\boldsymbol{\theta} \in \mathbb{R}^d} L(\boldsymbol{\theta})$.

### 3.2.1 Cost function for stationary probability function

Because the constant $c$ in Eq. (5) is unknown, for example, we cannot minimize a squared error such as $\sum_{x \in \mathcal{X}_o}(\pi^*(x) - \pi_\theta(x))^2$. Thus, we need to derive an alternative cost function of $\pi_\theta$ that is independent of $c$.

For $L_d(\boldsymbol{\theta})$, one natural choice might be a Kullback-Leibler (KL) divergence,

$$L_d^{\mathrm{KL}}(\boldsymbol{\theta}) \triangleq \sum_{x \in \mathcal{X}_o} \pi^*(x) \log \frac{\pi^*(x)}{\pi_\theta(x)} = -c \sum_{x \in \mathcal{X}_o} f(x) \log \pi_\theta(x) + o,$$

where $o$ is a term independent of $\boldsymbol{\theta}$. The minimizer of $L_d^{\mathrm{KL}}(\boldsymbol{\theta})$ is independent of $c$. However, minimization of $L_d^{\mathrm{KL}}$ will lead to a biased estimate. This is because $L_d^{\mathrm{KL}}$ will be decreased by increasing $\sum_{x \in \mathcal{X}_o} \pi_\theta(x)$ when the ratios $\pi_\theta(x)/\pi_\theta(x')$, $\forall x, x' \in \mathcal{X}_o$ are unchanged. This implies that, because of $\sum_{x \in \mathcal{X}_o} \pi_\theta(x) + \sum_{x \in (\mathcal{X} \backslash \mathcal{X}_o)} \pi_\theta(x) = 1$, minimizing $L_d^{\mathrm{KL}}$ has an unwanted side-effect of overvaluing $\sum_{x \in \mathcal{X}_o} \pi_\theta(x)$ and undervaluing $\sum_{x \in (\mathcal{X} \backslash \mathcal{X}_o)} \pi_\theta(x)$.

Here we propose an alternative form of $L_d$ that can avoid this side-effect. It uses a logarithmic ratio of the stationary probabilities such that

$$L_d(\boldsymbol{\theta}) \triangleq \frac{1}{2} \sum_{i \in \mathcal{X}_o} \sum_{j \in \mathcal{X}_o} \left( \log \frac{\pi^*(i)}{\pi^*(j)} - \log \frac{\pi_\theta(i)}{\pi_\theta(j)} \right)^2 = \frac{1}{2} \sum_{i \in \mathcal{X}_o} \sum_{j \in \mathcal{X}_o} \left( \log \frac{f(i)}{f(j)} - \log \frac{\pi_\theta(i)}{\pi_\theta(j)} \right)^2 \tag{8}$$

The log-ratio of probabilities represents difference of information contents between these probabilities in the sense of information theory [17]. Thus this function can be regarded as a sum of squared error between $\pi^*(x)$ and $\pi_\theta(x)$ over $x \in \mathcal{X}_o$ with respect to relative information contents. In a different point of view, Eq. (8) follows from maximizing the likelihood of $\boldsymbol{\theta}$ under the assumption that the observation "$\log f(i) - \log f(j)$" has a Gaussian white noise $\mathcal{N}(0, \epsilon^2)$. This assumption is satisfied when $f(i)$ has a log-normal distribution, $\mathcal{LN}(\mu_i, (\epsilon/\sqrt{2})^2)$, independently for each $i$, where $\mu_i$ is the true location parameter, and the median of $f(i)$ is equal to $e^{\mu_i}$.

### 3.2.2 Cost function for hitting probability function

Unlike $L_d(\boldsymbol{\theta})$, there are several options for $L_h(\boldsymbol{\theta})$. Examples of this cost function include a mean squared error and mean absolute error. Here we use the following standard squared errors in the log space, based on Eq. (6),

$$L_h(\boldsymbol{\theta}) \triangleq \frac{1}{2} \sum_{i \in \mathcal{X}_o} \sum_{j \in \mathcal{X}_o} \left( \log g(i, j) - \log h_\theta(j \mid i) \right)^2. \tag{9}$$

Eq. (9) follows from maximizing the likelihood of $\boldsymbol{\theta}$ under the assumption that the observation $\log g(i, j)$ has a Gaussian white noise, as with the case of $L_d(\boldsymbol{\theta})$.

## 4 Gradient-based Approach

Let us consider (local) minimization of the objective function $L(\boldsymbol{\theta})$ in Eq. (7). We adopt a gradient-descent approach for the problem, where the parameter $\boldsymbol{\theta}$ is optimized by the following iteration, with the notation $\nabla_{\boldsymbol{\theta}} L(\boldsymbol{\theta}) \triangleq [\partial L(\boldsymbol{\theta})/\partial \theta_1, \ldots, \partial L(\boldsymbol{\theta})/\partial \theta_d]^\top$,

$$\boldsymbol{\theta}_{t+1} = \boldsymbol{\theta}_t - \eta_t \boldsymbol{G}_{\theta_t}^{-1} \left\{ \gamma \nabla_{\boldsymbol{\theta}} L_d(\boldsymbol{\theta}_t) + (1-\gamma) \nabla_{\boldsymbol{\theta}} L_h(\boldsymbol{\theta}_t) + \lambda \nabla_{\boldsymbol{\theta}} R(\boldsymbol{\theta}_t) \right\}, \tag{10}$$

where $\eta_t > 0$ is an updating rate. The matrix $\boldsymbol{G}_{\theta_t} \in \mathbb{R}^{d \times d}$, called the metric of the parameter $\boldsymbol{\theta}$, is an arbitrary bounded positive definite matrix. When $\boldsymbol{G}_{\theta_t}$ is set to the identity matrix of size $d$, $\boldsymbol{I}_d$, the update formula in Eq. (10) becomes an ordinary gradient descent. However, since the tangent space at a point of a manifold representing $\mathrm{M}(\boldsymbol{\theta})$ is generally different from an orthonormal space with respect to $\boldsymbol{\theta}$ [4], one can apply the idea of natural gradient [3] to the metric $\boldsymbol{G}_\theta$, expecting to make the procedure more efficient. This is described in Section 4.1.

The gradients of $L_d$ and $L_h$ in Eq. (10) are given as

$$\nabla_{\boldsymbol{\theta}} L_d(\boldsymbol{\theta}) = \sum_{i \in \mathcal{X}_o} \sum_{j \in \mathcal{X}_o} \left( \log \frac{f(i)}{f(j)} - \log \frac{\pi_\theta(i)}{\pi_\theta(j)} \right) \left( \nabla_{\boldsymbol{\theta}} \log \pi_\theta(j) - \nabla_{\boldsymbol{\theta}} \log \pi_\theta(i) \right),$$

$$\nabla_{\boldsymbol{\theta}} L_h(\boldsymbol{\theta}) = \sum_{i \in \mathcal{X}_o} \sum_{j \in \mathcal{X}_o} \left( \log g(i,j) - \log h_\theta(j \mid i) \right) \nabla_{\boldsymbol{\theta}} \log h_\theta(j \mid i).$$

In order to implement the update rule of Eq. (10), we need to compute the gradient of the logarithmic stationary probability $\nabla_{\boldsymbol{\theta}} \log \pi_\theta$, the hitting probability $h_\theta$, and its gradient $\nabla_{\boldsymbol{\theta}} h_\theta$. In Sections 4.2, we will describe how to compute them, which will turn out to be quite non-trivial.

### 4.1 Natural gradient

Usually, a parametric family of Markov chains, $\mathcal{M}_\theta \triangleq \{ \mathrm{M}(\boldsymbol{\theta}) \mid \boldsymbol{\theta} \in \mathbb{R}^d \}$, forms a manifold structure with respect to the parameter $\boldsymbol{\theta}$ under information divergences such as a KL divergence, instead of the Euclidean structure. Thus the ordinary gradient, Eq. (10) with $\boldsymbol{G}_\theta = \boldsymbol{I}_d$, does not properly reflect the differences in the sensitivities and the correlations between the elements of $\boldsymbol{\theta}$. Accordingly, the ordinary gradient is generally different from the steepest direction on the manifold, and the optimization process with the ordinary gradient often becomes unstable or falls into a learning plateau [5].

For efficient learning, we consider an appropriate $\boldsymbol{G}_\theta$ based on the notion of the natural gradient (NG) [5]. The NG represents the steepest descent direction of a function $b(\boldsymbol{\theta})$ in a Riemannian space[3] by $-\boldsymbol{R}_\theta^{-1} \nabla_{\boldsymbol{\theta}} b(\boldsymbol{\theta})$ when the Riemannian space is defined by the metric matrix $\boldsymbol{R}_\theta$. An appropriate Riemannian metric on a statistical model, $Y$, having parameters, $\boldsymbol{\theta}$, is known to be its Fisher information matrix (FIM):[4]

$$\sum_y \Pr(Y=y \mid \boldsymbol{\theta}) \nabla_{\boldsymbol{\theta}} \log \Pr(Y=y \mid \boldsymbol{\theta}) \nabla_{\boldsymbol{\theta}} \log \Pr(Y=y \mid \boldsymbol{\theta})^\top.$$

In our case, the joint probability, $p_\theta(x'|x)\pi_\theta(x)$ for $x, x' \in \mathcal{X}$, fully specifies $\mathrm{M}(\boldsymbol{\theta})$ at the steady state, due to the Markovian property. Thus we propose to use the following $\boldsymbol{G}_\theta$ in the update rule of Eq. (10),

$$\boldsymbol{G}_\theta = \boldsymbol{F}_\theta + \sigma \boldsymbol{I}_d, \tag{11}$$

where $\boldsymbol{F}_\theta$ is the FIM of $p_\theta(x'|x)\pi_\theta(x)$,

$$\boldsymbol{F}_\theta \triangleq \sum_{x \in \mathcal{X}} \pi_\theta(x) \left( \nabla_{\boldsymbol{\theta}} \log \pi_\theta(x) \nabla_{\boldsymbol{\theta}} \log \pi_\theta(x)^\top + \sum_{x' \in \mathcal{X}} p_\theta(x'|x) \nabla_{\boldsymbol{\theta}} \log p_\theta(x'|x) \nabla_{\boldsymbol{\theta}} \log p_\theta(x'|x)^\top \right).$$

The second term with $\sigma \geq 0$ in Eq. (11) will be needed to make $\boldsymbol{G}_\theta$ positive definite.

$$\sum_y \Pr(Y=y \mid \boldsymbol{\theta}) \log \frac{\Pr(Y=y|\boldsymbol{\theta})}{\Pr(Y=y|\boldsymbol{\theta}+\Delta\boldsymbol{\theta})} \simeq \frac{1}{2} \|\Delta\boldsymbol{\theta}\|_{\boldsymbol{F}_\theta}^2.$$

## 4.2 Computing the gradient

To derive an expression for computing $\nabla_{\boldsymbol{\theta}} \log \pi_\theta$, we use the following notations for a vector and a matrix: $\boldsymbol{\pi}_\theta \triangleq [\pi_\theta(1), \ldots, \pi_\theta(|\mathcal{X}|)]^\top$ and $(\boldsymbol{P}_\theta)_{x,x'} \triangleq p_\theta(x'|x)$. Then the logarithmic stationary probability gradients with respect to $\boldsymbol{\theta}_i$ is given by

$$\frac{\partial}{\partial\theta_i}\log\boldsymbol{\pi}_\theta \triangleq \nabla_{\theta_i}\log\boldsymbol{\pi}_\theta = \mathrm{Diag}(\boldsymbol{\pi}_\theta)^{-1}(\boldsymbol{I}_d - \boldsymbol{P}_\theta^\top + \boldsymbol{\pi}_\theta\mathbf{1}_d^\top)^{-1}(\nabla_{\theta_i}\boldsymbol{P}_\theta^\top)\boldsymbol{\pi}_\theta, \tag{12}$$

where $\mathrm{Diag}(\boldsymbol{a})$ is a diagonal matrix whose diagonal elements consist of a vector $\boldsymbol{a}$, $\log\boldsymbol{a}$ is the element-wise logarithm of $\boldsymbol{a}$, and $\mathbf{1}_d$ denotes a column-vector of size $d$, whose elements are all 1. In the remainder of this section, we prove Eq. (12) by using the following proposition.

**Proposition 1 ([7])** *If* $\boldsymbol{A} \in \mathbb{R}^{d\times d}$ *satisfies* $\lim_{K\to\infty} A^K = \boldsymbol{0}$*, then the inverse of* $(\boldsymbol{I} - \boldsymbol{A})$ *exists, and* $(\boldsymbol{I} - \boldsymbol{A})^{-1} = \lim_{K\to\infty}\sum_{k=0}^K \boldsymbol{A}^k$.

Equation (3) is rewritten as $\boldsymbol{\pi}_\theta = \boldsymbol{P}_\theta^\top \boldsymbol{\pi}_\theta$. Note that $\boldsymbol{\pi}_\theta$ is equal to a normalized eigenvector of $\boldsymbol{P}_\theta^\top$ whose eigenvalue is 1. By taking a partial differential of Eq. (3) with respect to $\theta_i$, $\mathrm{Diag}(\boldsymbol{\pi}_\theta)\nabla_{\theta_i}\log\boldsymbol{\pi}_\theta = (\nabla_{\theta_i}\boldsymbol{P}_\theta^\top)\boldsymbol{\pi}_\theta + \boldsymbol{P}_\theta^\top\mathrm{Diag}(\boldsymbol{\pi}_\theta)\nabla_{\theta_i}\log\boldsymbol{\pi}_\theta$ is obtained. Though we get the following linear simultaneous equation of $\nabla_{\theta_i}\log\boldsymbol{\pi}_\theta$,

$$(\boldsymbol{I}_d - \boldsymbol{P}_\theta^\top)\mathrm{Diag}(\boldsymbol{\pi}_\theta)\nabla_{\theta_i}\log\boldsymbol{\pi}_\theta = (\nabla_{\theta_i}\boldsymbol{P}_\theta^\top)\boldsymbol{\pi}_\theta, \tag{13}$$

the inverse of $(\boldsymbol{I}_d - \boldsymbol{P}_\theta^\top)\mathrm{Diag}(\boldsymbol{\pi}_\theta)$ does not exist. It comes from the fact $(\boldsymbol{I}_d - \boldsymbol{P}_\theta^\top)\mathrm{Diag}(\boldsymbol{\pi}_\theta)\mathbf{1}_d = \boldsymbol{0}$. So we add a term including $\mathbf{1}_d^\top\mathrm{Diag}(\boldsymbol{\pi}_\theta)\nabla_{\theta_i}\log\boldsymbol{\pi}_\theta = \mathbf{1}_d^\top\nabla_{\theta_i}\boldsymbol{\pi}_\theta = \nabla_{\theta_i}\{\mathbf{1}_d^\top\boldsymbol{\pi}_\theta\} = 0$ to Eq. (13), such that $(\boldsymbol{I}_d - \boldsymbol{P}_\theta^\top + \boldsymbol{\pi}_\theta\mathbf{1}_d^\top)\mathrm{Diag}(\boldsymbol{\pi}_\theta)\nabla_{\theta_i}\log\boldsymbol{\pi}_\theta = (\nabla_{\theta_i}\boldsymbol{P}_\theta^\top)\boldsymbol{\pi}_\theta$. The inverse of $(\boldsymbol{I}_d - \boldsymbol{P}_\theta^\top + \boldsymbol{\pi}_\theta\mathbf{1}_d^\top)$ exists, because of Proposition 1 and the fact $\lim_{k\to\infty}(\boldsymbol{P}_\theta^\top - \boldsymbol{\pi}_\theta\mathbf{1}_d^\top)^k = \lim_{k\to\infty}\boldsymbol{P}_\theta^{\top k} - \boldsymbol{\pi}_\theta\mathbf{1}_d^\top = \boldsymbol{0}$. The inverse of $\mathrm{Diag}(\boldsymbol{\pi}_\theta)$ also exists, because $\pi_\theta(x)$ is positive for any $x \in \mathcal{X}$ under Assumption 1. Hence we get Eq. (12).

To derive expressions for computing $h_\theta$ and $\nabla_{\boldsymbol{\theta}}\log h_\theta$, we use the following notations: $\boldsymbol{h}_\theta(x) \triangleq [h_\theta(x\,|\,1), \ldots, h_\theta(x\,|\,|\mathcal{X}|)]^\top$ for the hitting probabilities in Eq. (4) and $(\boldsymbol{P}_{\mathrm{T}\theta})_{x,x'} \triangleq p_{\mathrm{T}\omega}(x'\,|\,x)$ for *p-transition* probabilities in Eq. (1). The hitting probabilities and those gradients with respect to $\theta_i$ can be computed as the following closed forms,

$$\boldsymbol{h}_\theta(x) = (\boldsymbol{I}_{|\mathcal{X}|} - \beta\boldsymbol{P}_{\mathrm{T}\theta}^{\setminus x})^{-1}\boldsymbol{e}_{|\mathcal{X}|}^x, \tag{14}$$

$$\nabla_{\theta_i}\log\boldsymbol{h}_\theta(x) = \beta\,\mathrm{Diag}(\boldsymbol{h}_\theta(x))^{-1}(\boldsymbol{I}_{|\mathcal{X}|} - \beta\boldsymbol{P}_{\mathrm{T}\theta}^{\setminus x})^{-1}(\nabla_{\theta_i}\boldsymbol{P}_\theta^{\setminus x})\boldsymbol{h}_\theta(x), \tag{15}$$

where $\boldsymbol{e}_{|\mathcal{X}|}^x$ denotes a column-vector of size $|\mathcal{X}|$, where $x$'th element is 1 and all of the other elements are zero. The matrix $\boldsymbol{P}_{\mathrm{T}\theta}^{\setminus x}$ is defined as $(\boldsymbol{I}_{|\mathcal{X}|} - \boldsymbol{e}_{|\mathcal{X}|}^x\boldsymbol{e}_{|\mathcal{X}|}^{x\top})\boldsymbol{P}_{\mathrm{T}\theta}$. We will derive Eqs. (14) and (15) as follows. The hitting probabilities in Eq. (4) can be represented as the following recursive form,

$$h_\theta(x'\,|\,x) = \begin{cases} 1 & \text{if } x' = x \\ \beta\sum_{y\in\mathcal{X}}p_{\mathrm{T}\omega}(y\,|\,x)\,h_\theta(x'\,|\,y) & \text{otherwise.} \end{cases}$$

This equation can be represented with the matrix notation as $\boldsymbol{h}_\theta(x) = \boldsymbol{e}_{|\mathcal{X}|}^x + \beta\boldsymbol{P}_{\mathrm{T}\theta}^{\setminus x}\boldsymbol{h}_\theta(x)$. Because the inverse of $(\boldsymbol{I}_{|\mathcal{X}|} - \beta\boldsymbol{P}_{\mathrm{T}\theta}^{\setminus x})$ exists by Proposition 1 and $\lim_{k\to\infty}(\beta\boldsymbol{P}_{\mathrm{T}\theta}^{\setminus x})^k = \boldsymbol{0}$, we get Eq. (14). In a similar way, one can prove Eq. (15).

## 5 Implementation

For implementing the proposed method, parametric models of the initial probability $p_{\mathrm{I}\nu}$ and the *p-transition* probability $p_{\mathrm{T}\omega}$ in Eq. (1) need to be specified. We provide intuitive models based on the logit function [8].

The initial probability is modeled as

$$p_{\mathrm{I}\nu}(x) \triangleq \frac{\exp(s_{\mathrm{I}}(x;\boldsymbol{\nu}))}{\sum_{y\in\mathcal{X}}\exp(s_{\mathrm{I}}(y;\boldsymbol{\nu}))}, \tag{16}$$

where $s_{\mathrm{I}}(x;\boldsymbol{\nu})$ is a state score function with its parameter $\boldsymbol{\nu} \triangleq [\boldsymbol{\nu}^{\mathrm{loc}\top}, \boldsymbol{\nu}^{\mathrm{glo}\top}]^\top \in \mathbb{R}^{d_1}$ consisting of a local parameter $\boldsymbol{\nu}^{\mathrm{loc}} \in \mathbb{R}^{|\mathcal{X}|}$ and a global parameter $\boldsymbol{\nu}^{\mathrm{glo}} \in \mathbb{R}^{d_1-|\mathcal{X}|}$. It is defined as

$$s_{\mathrm{I}}(x;\boldsymbol{\nu}) \triangleq \nu_x^{\mathrm{loc}} + \boldsymbol{\phi}_{\mathrm{I}}(x)^\top\boldsymbol{\nu}^{\mathrm{glo}}, \tag{17}$$

where $\phi_I(x) \in \mathbb{R}^{d_1 - |\mathcal{X}|}$ is a feature vector of a state $x$. In the case of the road network, a state corresponds to a road segment. Then $\phi_I(x)$ may, for example [18], be defined with the indicators of whether there are particular types of buildings near the road segment, $x$. We refer to the first term and the second term of the right-hand side in Eq. (17) as a local preference and a global preference, respectively. If a simpler model is preferred, either of them would be omitted.

Similarly, a *p-transition* probability model with the parameter $\boldsymbol{\omega} \triangleq [\boldsymbol{\omega}^{\mathrm{loc}\top}, \boldsymbol{\omega}_1^{\mathrm{glo}\top}, \boldsymbol{\omega}_2^{\mathrm{glo}\top}]^\top$ is given as

$$p_{\mathrm{T}\omega}(x'|x) \triangleq \begin{cases} \exp(s_{\mathrm{T}}(x, x'; \boldsymbol{\omega}))\big/ \sum_{y \in \mathcal{X}_x} \exp(s_{\mathrm{T}}(x, y; \boldsymbol{\omega})), & \text{if } (x, x') \in \mathcal{X} \times \mathcal{X}_x, \\ 0 & \text{otherwise,} \end{cases} \quad (18)$$

where $\mathcal{X}_x$ is a set of states connected from $x$, and $s_{\mathrm{T}}(x, x'; \boldsymbol{\omega})$ is a state-to-state score function. It is defined as

$$s_{\mathrm{T}}(x, x'; \boldsymbol{\omega}) \triangleq \omega_{(x,x')}^{\mathrm{loc}} + \boldsymbol{\phi}_{\mathrm{T}}(x')^\top \boldsymbol{\omega}_1^{\mathrm{glo}} + \boldsymbol{\psi}(x, x')^\top \boldsymbol{\omega}_2^{\mathrm{glo}}, \quad (x, x') \in \mathcal{X} \times \mathcal{X}_x,$$

where $\omega_{(x,x')}^{\mathrm{loc}}$ is the element of $\boldsymbol{\omega}^{\mathrm{loc}}$ ($\in \mathbb{R}^{\sum_{x \in \mathcal{X}} |\mathcal{X}_x|}$) corresponding to transition from $x$ to $x'$, and $\boldsymbol{\phi}_{\mathrm{T}}(x)$ and $\boldsymbol{\psi}(x, x')$ are feature vectors. For the road network, $\boldsymbol{\phi}_{\mathrm{T}}(x)$ may be defined based on the type of the road segment, $x$, and $\boldsymbol{\psi}(x, x')$ may be defined based on the angle between $x$ and $x'$. Those linear combinations with the global parameters, $\boldsymbol{\omega}_1^{\mathrm{glo}}$ and $\boldsymbol{\omega}_2^{\mathrm{glo}}$, can represent drivers' preferences such as how much the drivers prefer major roads or straight routes to others.

Note that the $p_{I\nu}(x)$ and $p_{\mathrm{T}\omega}(x'|x)$ presented in this section can be differentiated analytically. Hence, $\boldsymbol{F}_\theta$ in Eq. (11), $\nabla_{\theta_i} \log \boldsymbol{\pi}_\theta$ in Eq. (12), and $\nabla_{\theta_i} \boldsymbol{h}_\theta$ in Eq. (15) can be computed efficiently.

## 6 Experiments

### 6.1 Experiment on synthetic data

To study the sensitivities of the performance of our algorithm to the ratio of observable states, we applied it to randomly synthesized inverse problems of 100-state Markov chains with a varying number of observable states, $|\mathcal{X}_{\mathrm{o}}| \in \{5, 10, 20, 35, 50, 70, 90\}$. The linkages between states were randomly generated in the same way as [19]. The values of $p_I$ and $p_T$ are determined in two stages. First, the basic initial probabilities, $p_{I\nu}$, and the basic transition probabilities, $p_{\mathrm{T}\omega}$, were determined based on Eqs. (16) and (18), where every element of $\boldsymbol{\nu}, \boldsymbol{\omega}, \boldsymbol{\phi}_I(x), \boldsymbol{\phi}_{\mathrm{T}}(x)$, and $\boldsymbol{\psi}_{\mathrm{T}}(x, x')$ was drawn independently from the normal distribution $\mathcal{N}(0, 1^2)$. Then we added noises to $p_{I\nu}$ and $p_{\mathrm{T}\omega}$, which are ideal for our algorithm, by using the Dirichlet distribution $\mathtt{Dir}$, such that $p_I = 0.7 p_{I\nu} + 0.3\boldsymbol{\sigma}$ with $\boldsymbol{\sigma} \sim \mathtt{Dir}(0.3 \times \mathbf{1}_{|\mathcal{X}|})$. Then we sampled the visiting frequencies $f(x)$ and the hitting rates $g(x, x')$ for every $x, x' \in \mathcal{X}_{\mathrm{o}}$ from this synthesized Markov chain.

We used Eqs. (16) and (18) for the models and Eq. (7) for the objective of our method. In Eq. (7), we set $\gamma = 0.1$ and $R(\boldsymbol{\theta}) = \|\boldsymbol{\theta}\|_2^2$, and $\lambda$ was determined with a cross-validation. We evaluated the quality of our solution with the relative mean absolute error (RMAE), $\mathrm{RMAE} = \frac{1}{|\mathcal{X} \backslash \mathcal{X}_{\mathrm{o}}|} \sum_{x \in \mathcal{X} \backslash \mathcal{X}_{\mathrm{o}}} \frac{|f(x) - \hat{c}\pi_\theta(x)|}{\max\{f(x), 1\}}$, where $\hat{c}$ is a scaling value given by $\hat{c} = 1/|\mathcal{X}_{\mathrm{o}}| \sum_{x \in \mathcal{X}_{\mathrm{o}}} f(x)$. As a baseline method, we use Nadaraya-Watson kernel regression (NWKR) [8] whose kernel is computed based on the number of hops in the minimum path between two states. Note that the NWKR could not use $g(x, x')$ as an input, because this is a regression problem of $f(x)$. Hence, for a fair comparison, we also applied a variant of our method that does not use $g(x, x')$.

Figure 2 (A) shows the mean and standard deviation of the RMAEs. The proposed method gives clearly better performance than the NWKR. This is mainly due to the fact that the NWKR assumes that all propagations of the observation from a link to another connected link are equally weighted. In contrast, our method incorporates such weight in the transition probabilities.

### 6.2 Experiment on real-world traffic data

We tested our method through a city-wide traffic-monitoring task as shown in Fig. 1. The goal is to estimate the traffic volume along an arbitrary road segment (or link of a network), given observed traffic volumes on a limited number of the links, where a link corresponds to the state $x$ of $\mathrm{M}(\boldsymbol{\theta})$, and the traffic volume along $x$ corresponds to $f(x)$ of Eq. (5). The traffic volumes along the observable links were reliably estimated from real-world web-camera images captured in Nairobi, Kenya [2,

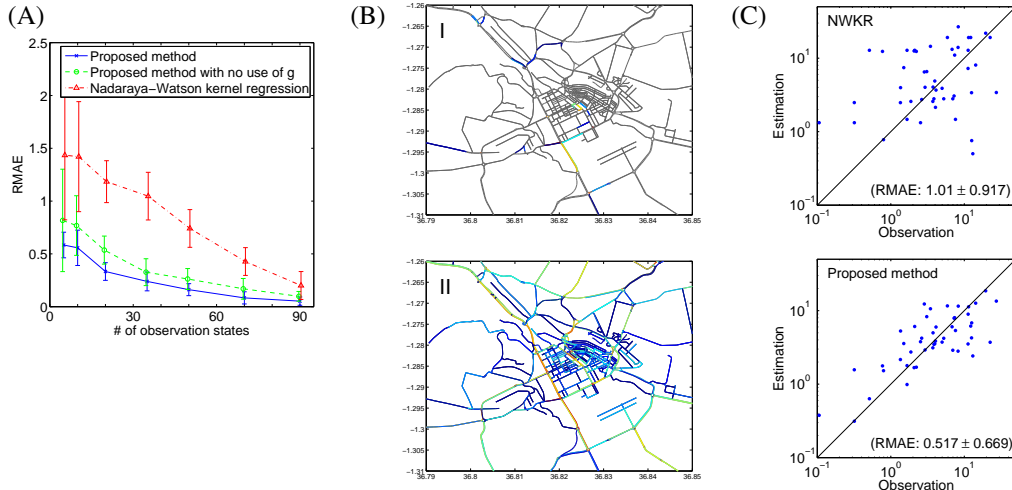

Figure 2: (A) Comparison of RMAE for the synthetic task between our methods and the NWKR (baseline method). (B) Traffic volumes for a city center map in Nairobi, Kenya, I: Web-camera observations (colored), II: Estimated traffic volumes by our method. (C) Comparison between the NWKR and our method for the real traffic-volume prediction problem.

15], while we did not use the hitting rate $g(x, x')$ here because of its unavailability. Note that this task is similar to network tomography [27, 30] or link-cost prediction [32, 14]. However, unlike network tomography, we need to infer all of the link traffics instead of source-destination demands. Unlike link-cost prediction, our inputs are stationary observations instead of trajectories. Again, we use the NMKR as the baseline method. The road network and the web-camera observations are shown in Fig. 2 (B)-I. While the total number of links was $1,497$, the number of links with observations was only 52 (about 3.5%). We used the parametric models in Section 5, where $\phi_T(x) \in [-1, 1]$ was set based on the road category of $x$ such that primary roads have a higher value than secondary roads [22], and $\psi(x, x') \in [-1, 1]$ was the cosine of the angle between $x$ and $x'$. However, we omitted the terms of $\phi_I(x)$ in Eq. (17).

Figure 2 (B)-II shows an example of our results, where the red and yellow roads are most congested while the traffic on the blue roads is flowing smoothly. The congested roads from our analysis are consistent with those from a local traffic survey report [13]. Figure 2 (C) shows comparison between predicted and observed travel volumes. In the figures, the $45°$ line corresponds to perfect agreement between the actual and predicted values. To evaluate accuracy, we employed the leave-one-out cross-validation. We can see that the proposed method gives a good performance. This is rather surprising, because the rate of observation links is very limited to only $3.5$ percent.

## 7   Conclusion

We have defined a novel inverse problem of a Markov chain, where we infer the probabilities about the initial states and the transitions, using a limited amount of information that we can obtain by observing the Markov chain at a small number of states. We have proposed an effective objective function for this problem as well as an algorithm based on natural gradient.

Using real-world data, we have demonstrated that our approach is useful for a traffic monitoring system that monitors the traffic volume at limited number of locations. From this observation the Markov chain model is inferred, which in turn can be used to deduce the traffic volume at any location. Surprisingly, even when the observations are made at only several percent of the locations, the proposed method can successfully infer the traffic volume at unobserved locations.

Further analysis of the proposed method is necessary to better understand its property and effectiveness. In particular, our future work includes an analysis of model identifiability and empirical studies with other applications, such as logistics and economic system modeling.

**Acknowledgments**
The authors thank Dr. R. Morris, Dr. R. Raymond, and Mr. T. Katsuki for fruitful discussion.

## Footnotes

[1] The rate $\beta$ can depend on the current state $x$ so that $\beta$ can be replaced with $\beta(x)$ throughout the paper. For readability, we assume $\beta$ is a constant.

[2]We assume $\frac{\partial}{\partial \theta_i} \log p_{I\nu}(x) = 0$ when $p_{I\nu}(x) = 0$, and an analogous assumption applies to $p_{T\omega}$.

[3] A parameter space is a Riemannian space if the parameter $\boldsymbol{\theta} \in \mathbb{R}^d$ is on a Riemannian manifold defined by a positive definite matrix called a Riemannian metric matrix $\boldsymbol{R}_\theta \in \mathbb{R}^{d \times d}$. The squared length of a small incremental vector $\Delta\boldsymbol{\theta}$ connecting $\boldsymbol{\theta}$ to $\boldsymbol{\theta} + \Delta\boldsymbol{\theta}$ in a Riemannian space is given by $\|\Delta\boldsymbol{\theta}\|_{\boldsymbol{R}_\theta}^2 = \Delta\boldsymbol{\theta}^\top \boldsymbol{R}_\theta \Delta\boldsymbol{\theta}$.

[4] The FIM is the unique metric matrix of the second-order Taylor expansion of the KL divergence, that is,

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
