[Reviews · NeurIPS 2013]

Submitted by Assigned_Reviewer_4

The paper proposes an estimation strategy for recovering the parameters of a finite state Markov chain given observed stationary frequencies of some states. (The paper also considers using additional hitting frequencies for a few state pairs, but this is not used in the final evaluation.)

Although the problem proposed is potentially very interesting, the paper does not appear to be in a mature state. Some fundamental issues are not adequately addressed, while the evaluation is limited, and the writing quality is not strong.

Note that there is an uncountable set of ergodic transition models that can exactly match a given subset of stationary frequencies when the number of observed stationary state frequencies is small relative to the total number of states. In short, the problem is ill posed. This means that additional prior knowledge is required to recover a unique transition model, whether via regularization or imposing a compact parametric form.

Unfortunately, the paper does not even discuss the necessity of imposing prior knowledge to achieve non-vacuous results in this setting. That is, even though the paper proposes a parametric representation and considers regularization, it does not observe that the problem is ill posed, hence that *zero* loss is obtainable for an uncountable set of transition models. Instead, the paper merely proposes a few heuristic loss functions and an arbitrary regularizer. This seems to be a somewhat superficial approach to the issue. Even the paper's main motivating example of a traffic network possesses clear prior structure: the transition probability must be zero between unconnected links. There will be some subsets of stationary state probabilities that can identify the transition model in such a structured setting, but this possibility is not even discussed.

The proposed loss functions (8) and (9) are clearly heuristic, and I found the attempted justification below (8) to be unconvincing. A distributional assumption can always be reverse engineered from a loss formulation like (8), but that does not mean the log-frequencies will have this distributional form.

The discussion of "bias" in Section 3.2 appears to be too informal: on the context of estimation, unbiasedness normally refers to a property that the expected value of an estimator is the true value of the underlying parameter. Although the loss formulation (8) is claimed to be "unbiased", no argument is given that the expected value of the minimizer of (8) has the correct value.

If one accepts the regularized loss formulation, the natural gradient approach developed in Section 4 appears to be reasonable. The technical development in this section is sound.

The experimental evaluation appears to be limited. One synthetic and one "real" experiment are conducted.

In the synthetic experiment, the model exactly matches that used by the paper's proposed estimator, which is an ideal scenario that cannot be guaranteed in practice. Here the proposed method's results are compared to a baseline estimator (NWKR) that does not exploit the fact that the target stationary frequencies are associated with a Markov chain. This seems to be a weak straw-man competitor, and it is not too surprising that it performs poorly by comparison.

In the real experiment, it is somewhat disappointing that hitting rate observations are not included in the observations, which means that the initial distribution is irrelevant to the estimation problem (since only stationary frequencies are being used). Nevertheless, this is an interesting experimental domain. The results look reasonable, although again, the baseline competitor is too weak to take very seriously. A more credible competitor would identify a transition model that exactly matches the observed stationary frequencies (and the sparsity pattern implied by the road network) while maximizing entropy. The experimental evaluation would also be more convincing if results could be obtained for a larger number of synthetic and real domains.
Summary: The paper considers and interesting problem: estimating the parameters for a Markov chain from observed stationary frequencies of a subset of states. Unfortunately, the proposed estimator, based on a regularized loss minimization, is heuristic and needs a stronger justification. The experimental evaluation is interesting but brief.

Submitted by Assigned_Reviewer_5

This paper addresses a novel and interesting problem in Markov chain - finding transition probabilities from partial observations with a limited number of states observable. The problem is well motivated and formulated. A gradient based optimization method is proposed to address the problem.

The experiment section could have been improved. It would be interesting to see more analysis of the experiment results; for example a quantitative evaluation of Fig.2(B) if it is possible.
Summary: This paper proposes to addresses a new and interesting problem. The paper is well written.

Submitted by Assigned_Reviewer_6

This paper addresses the problem of learning the parameters of a Markov model (not necessarily as a table, but possibly parameterized in some arbitrary way), based upon partial observations of state visitation frequencies (some states may not be observed at all) and pairwise observations of states that are visited sequentially.

The authors describe the problem well, making a reasonable case that version of the problem is different from others yet well motivated by practical concerns (I may be unaware of some work in this area, so I need to take the authors' word for this), and propose a reasonable solution to the problem. The paper is generally clear and well written, and the results seem good, though lack of ground truth data for the real world problem make this a little difficult to assess fully.

I expect some Bayesians might complain that (fill in your favorite Bayesian dogma here), but I can live with this.

Overall, I think this is a solid contribution.

Minor comments:

I found the use of X_0 and caligraphic X_0 to be a bit confusing because they look very similar but mean quite different things.

Line 186: It looks like you have a forall in superscript.

Line 190: independent c -> independent of c

The summation in (8) is a bit confusing. How can a state be larger than another state? I'm assuming that you have imposed an ordering on the states and that you are abusing notation a bit here to use the states themselves in places where you might use the indices of the states?

Line 415: percents -> percent

Summary: This is a nice approach to a variation on the problem of inferring the parameters of a Markov model from partial information.
Author Feedback

Author rebuttal: We very much appreciate the careful reviews, valuable comments, and also the suggestions for our paper. Our paper will be much better with revisions based on these comments.

We will focus on the most critical misunderstanding of the first reviewer (Assigned_Reviewer_4) about regularization and identifiability. This reviewer states “[the paper] does not observe that the problem is ill posed, hence that *zero* loss is obtainable for an uncountable set of transition models. ... a traffic network possesses clear prior structure: the transition probability must be zero between unconnected links. There will be some subsets of stationary state probabilities that can identify the transition model in such a structured setting, but this possibility is not even discussed.”

Although it is not explicitly stated in the original manuscript, we generally assume that a network structure is known or roughly estimated. As the reviewer suggests, network structures or connectivity will be available in many applications such as road networks. So, we does use this prior knowledge of the network connectivity structure in the transition model of Eq. (18). In Eq. (18), transition probabilities between unconnected links are defined as zero. We believe that, when an average degree of a graph is (much) less than the number of states, the problem is not strongly ill-posed and then we can achieve non-vacuous results by the use of our regularization, as is presented in the experimental results. The theoretical analysis of this identifiability is our important future work.

We would like to carefully add those explanations and discussions in the final version.